# DATA-FREE VFX SELF-MINING

## ABSTRACT

We present AutoVFX, an automated framework for extracting and amplifying visual-effects (VFX) capabilities from pretrained Image-to-Video (I2V) foundation models, thereby obviating costly manual dataset construction and annotation. Motivated by the observation that contemporary I2V models possess latent but unreliable VFX competence, we operationalizes a closed-loop agent composed of four coordinated modules: *i)* VFX Designer: structured prompt exploration and decomposition via an LLM; *ii)* Scene Artist: VFX-aware first-frame synthesis using state-of-the-art text-to-image models and automated image selection; *iii)* Video Producer: I2V synthesis with multimodal per-clip evaluation (perceptual quality metrics and semantic consistency); and *iv)* VFX Refiner: selective data curation and cycle-finetuning of the I2V backbone. Central to our approach is a scalable multimodal quality controller that enforces both per-frame aesthetic fidelity and per-clip semantic alignment, and a cycle-finetuning regime that iteratively improves training data and model behavior. To assess performance, we introduce VFX-Bench, a diverse suite of challenging VFX tasks, and report two complementary metrics termed Comprehensive Score and Success Rate. Empirical evaluation demonstrates that AutoVFX substantially raises performance relative to off-the-shelf I2V baselines, yields favorable scalability and cost profiles compared to manual dataset approaches, and outperforms several VFX-tailored baselines. All data and code will be made publicly available.

## 1 INTRODUCTION

Recent advances in video generation models have significantly expanded the capabilities of synthesized videos. From the proprietary Veo 3 (Google Deep-Mind) model to community-driven models such as Hun-yuanVideo (Kong et al., 2024), Open-Sora (Peng et al., 2025), and Wan2.1 (Wan et al., 2025), video generation has become increasingly realistic, especially the newly released Wan2.2 model (Wan et al., 2025). This progress has unlocked strong potential for many downstream applications (Che et al., 2024; Liu et al., 2025a; Wu et al., 2025), notably visual effects

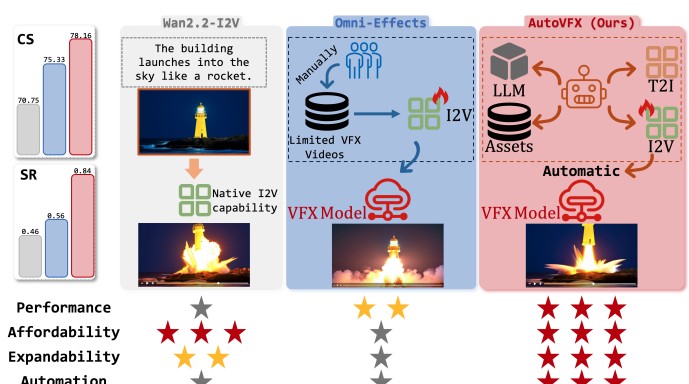

Figure 1: **Comparison with current open-source Models with VFX capabilities.** Thanks to the design of the automated agent, AutoVFX achieves the highest metrics (CS and SR) while featuring lower costs and better scalability. Quantitative results are derived from Tab. 2

(VFX), where generative models can create stylized or physically implausible phenomena that are costly or impossible to produce with traditional techniques. However, two key limitations block practical adoption of these models for VFX. *First*, off-the-shelf video generation models exhibit poor VFX-specific generalization and low success rates when asked to produce complex effects (see Fig. 1-Left). *Second*, the obvious remedy that builds VFX-centric datasets and fine-tuning models (*e.g.*, Omni-Effects) is expensive and slow because it depends on manual data collection and expert

annotation (see Fig. 1-Middle). At the same time, we observe that naive image-to-video models already sometimes produce convincing effects under carefully designed prompts, indicating latent VFX capability that is not being fully exploited.

Motivated by these observations, we propose a practical, automated path to unlock VFX performance from pretrained video generation models. Our method, called AutoVFX, replaces costly manual processes with a lightweight visual-effect agent that coordinates multiple tools and roles to: *1)* automatically generate candidate VFX videos from foundation models; *2)* evaluate and filter outputs using a multimodal quality controller that inspects everything from the first frame to the whole clip; and *3)* iteratively fine-tune the foundation model in cycles to progressively improve training data and generation fidelity. By closing the loop between generation, multimodal evaluation, and cycle-finetuning, AutoVFX yields a customized VFX model with high success rates and low human cost (see Fig. 1-Right).

- A novel AutoVFX agent framework that automatically mines the VFX potential from pretrained video generation models, achieving automation, low-cost, and high-efficiency across the entire procedure.

- A scalable multimodal evaluation module to enforce per-frame and per-VFX-video quality, together with a cycle-finetuning strategy that iteratively improves data and model quality to fully mine the VFX potential of the I2V foundation models..

- Empirical results on the proposed VFX-Bench, which covers a diverse range of trending visual effects, show that AutoVFX can effectively mine the potential and substantially raises the performance of the I2V model while remaining far more scalable and cost-efficient than manual VFX dataset approaches, even surpassing VFX-tailored models to achieve state-of-the-art performance.

## 2 AUTOVFX: AUTOMATIC VFX CREATION

### 2.1 VISUAL EFFECT AGENT: MINING VFX POTENTIAL FOR PRETRAINED I2V MODEL

Image-to-Video models like Wan2.2-I2V (Wan et al., 2025) exhibit certain ability to generate VFX content, but the quality of their generation is often inconsistent. While they can generate visual effects in various styles, the performance can vary significantly depending on the complexity of the visual effects. Therfore, the goal of AutoVFX is to automatically generate high-quality VFX videos by leveraging pretrained I2V models and mining their VFX potential through iterative fine-tuning.

The core of this process is driven by a specialized VFX Agent, which orchestrates the collaborative efforts of multiple intelligent components. These intelligent components work together to guide the VFX generation process, continuously finetune the pretrained I2V model and improving its ability to produce high-quality VFX videos. Based on the different responsibilities assumed by each component within the VFX Agent, we personify these components into four role tools:

- **VFX Designer** is responsible for obtaining the desired visual effects and converting them into a format that is more understandable and interpretable by the I2V model.

- **Scene Artist** generates the first-frame of the VFX video, which serves as the foundation for the entire video sequence.

- **Video Producer** combines inputs from the VFX Designer and Scene Artist, and transforms them into a professionally executed VFX video, ensuring that the final product meets the creative vision for practical application.

- **VFX Refiner** carefully selects the best-performing videos and uses them to finetune the I2V model through iterative feedback, ensuring continuous improvement in the quality and stability of the generated VFX.

As shown in Fig. 2, VFX Agent begins the process by assigning the VFX Designer the task of handling the user's VFX inputs or autonomously searching for trending VFX ideas. Once these VFX effects are refined and tailored to the needs of the project, the Designer passes them on to the Scene Artist. The Artist, using the refined VFX prompts, generates a range of first-frame images in various styles that will serve as the foundation for the entire VFX video. Next, the Video Producer

takes these carefully curated VFX and first frames, and transforms them into a fully realized VFX video. The Producer then conducts an automated evaluation of the video's visual quality, ensuring that the generated VFX meet the desired standards. The generated videos are then handed off to the VFX Refiner, who applies a selective strategy to pick the best-performing outputs. These high-quality videos are used to fine-tune the I2V model, improving its ability to generate better VFX over time. It's important to note that the VFX Agent does not end the process here. Instead, it allows the Video Producer to use the updated I2V model to generate even higher-quality VFX videos. This iterative feedback loop continues, with the VFX Refiner selecting the best-performing videos after each round and feeding them back into the I2V model for further fine-tuning. Through this continuous, collaborative process, the VFX Agent ensures that the I2V model's VFX potential is fully mined and enhanced, gradually generating stable and high-quality VFX videos.

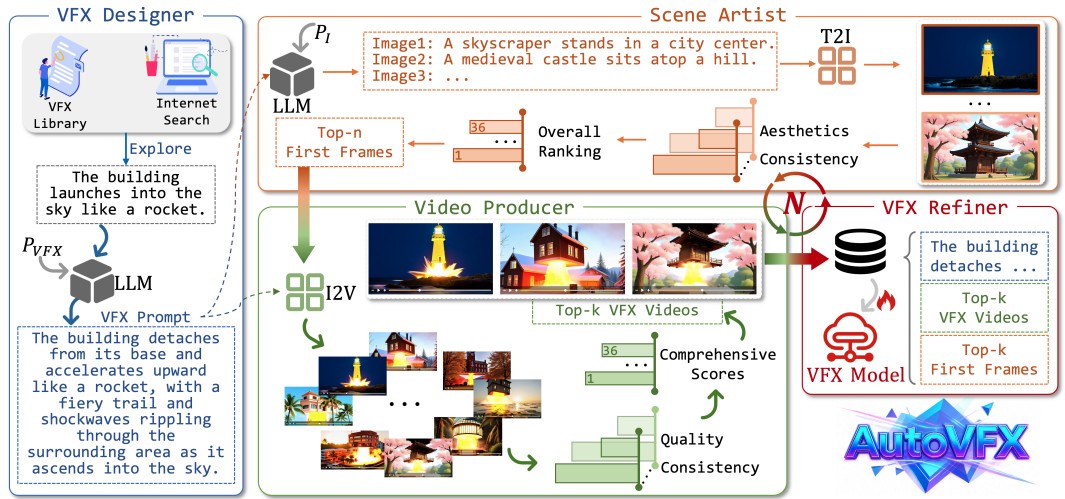

Figure 2: **Overview of AutoVFX** that consists of: *1)* **VFX Designer** (Sec. 2.2) the content of visual effects; *2)* **Scene Artist** (Sec. 2.3) constructs the relevant initial frame images; *3)* **Video Producer** (Sec. 2.4) generates and automatically filters high-quality initial VFX videos; *4)* **VFX Refiner** (Sec. 2.5) fine-tunes the basic I2V model with filtered data for self-mining the potential of visual effects. After $N$ iterations of automation, the final VFX model is obtained for VFX applications.

## 2.2 VFX DESIGNER: PROMPT EXPLORATION AND CRAFTING

As the initiator of the AutoVFX, the **VFX Designer** is envisioned as the role responsible for shaping creative intent into actionable design instructions, much like a designer in a production pipeline who bridges abstract ideas with concrete implementations.

**VFX Prompt Exploration.** At the core of the VFX Designer lies a Large Language Model (LLM) (Yang et al., 2025), which drives the exploration of diverse VFX ideas. In the main procedure, users are guided to explore a curated VFX library whose effects are characterized by diversity, timeliness, and creativity, enabling them to select visual effects that best match their intent. Beyond this, the Designer can also autonomously search the web for trending VFX that reflect contemporary aesthetics. Through these pathways, the VFX Designer expands the space of exploration, uncovering novel and engaging VFX playstyles that serve as the foundation for subsequent creation.

**VFX Prompt Crafting.** Naive VFX descriptions are often coarse and abstract, typically capturing only high-level intentions such as "make it fly in the air" (Mao et al., 2025). To enable controllable and high-quality generation, these broad descriptions must be decomposed into fine-grained elements, including the scene, the main subject, and the stylistic attributes. At this stage, the VFX Designer leverages the reasoning ability of the LLMs to perform this refinement, transforming vague user inputs or automatically discovered effects into structured prompts that are more interpretable for the I2V model with prompt $P_{VFX}$. This process provides precise and detailed guidance, ensuring that the generative model can capture both the global intent and the nuanced details necessary for producing visually coherent and controllable VFX videos.

## 2.3 SCENE ARTIST: VFX-SPECIFIC SCENE CREATION

As an architect within the AutoVFX procedure, the **Scene Artist** is responsible for transforming abstract creative concepts into concrete visual elements. Much like an artist who takes a script and brings it to life through visual storytelling, the Scene Artist ensures that the first-frame image not only aligns with the VFX prompt but also serves as the foundational visual anchor for the AutoVFX, setting the tone and narrative direction for the video.

**Text-to-Image Model Selection** The first frame serves as the cornerstone of the VFX generation, establishing the tone and style of the scene upon which subsequent frames are built. As a result, the choice of Text-to-Image (T2I) model for generating the first frame is crucial, since the fidelity, style, and semantic accuracy of the first frame directly affect the overall quality of the final VFX. A variety of advanced image generation models are available for the task, including Stable Diffusion (Rombach et al., 2022), DALL·E (Ramesh et al., 2021), Imagen (Saharia et al., 2022), and the recent FLUX family (Labs et al., 2025). To ensure flexibility, the VFX Agent is designed with an extensible architecture that allows multiple T2I tools to be integrated and selected according to the requirements of different tasks. Specifically, we adopt the FLUX series of models as the default tools. This choice is motivated by its ability to produce high-quality images with consistent structural coherence, as well as its strong semantic alignment with input prompts and controllable stylistic variation. In addition, its dual capability in image generation and image editing makes it naturally compatible with our branch for first-last-frame video generation. These properties render FLUX particularly suitable for VFX-specific scene creation, where the generated first frame is required to exhibit both stylistic diversity and detailed precision in order to reliably guide subsequent video synthesis.

**First-frame Creation.** The quality of the first frame in VFX video generation is largely determined by the image prompt that drives the T2I model. To exert fine-grained control over this process, we employ an LLM to transform the crafted VFX prompt into a diverse set of detailed image prompts with prompt $P_I$. Each image prompt specifies the subject and surrounding environment in a static configuration, while introducing stylistic variation through factors such as background, atmosphere, or lighting. These prompts serve as precise instructions that guide the T2I model in producing candidate first-frame images tailored to the intended VFX. A practical challenge arises in balancing prompt quantity with diversity and fidelity. Although generating a very large number of prompts could increase variation, the token limitations of LLMs often degrade quality and lead to repetition. To address this, we constrain the set to 100 prompts and instead leverage the stochasticity of the FLUX sampling process: by varying random seeds, multiple high-quality and diverse image candidates can be obtained from the same prompt, thereby maintaining both efficiency and prompt fidelity. Beyond this procedure, we also support an optional last-frame editing branch. Here, a Multimodal Large Language Model(MLLM) (Bai et al., 2025; Zhu et al., 2025; Hong et al., 2025) imagines the final state of the effect and produces a descriptive prompt for the last frame, which is realized by the image editing capability of FLUX. While this branch offers additional controllability, it introduces extra complexity that poses greater challenges to image editing models. Consequently, our main pipeline adopts first-frame guidance as the default strategy, ensuring reliability and coherence in subsequent VFX generation.

**Automatic Image Assessment.** Since the quality of the first frame critically affects the effectiveness of subsequent VFX generation, we design an automatic image evaluation module to select the most suitable candidates from the diverse set of generated images. This evaluation framework incorporates two complementary dimensions: *1)* **image aesthetic scoring** (Schuhmann, 2022) measures the overall visual appeal of the image. Beyond simple heuristics, this scoring considers factors such as composition, color harmony, clarity, and contrast, thereby reflecting human-perceived aesthetic quality. By prioritizing aesthetically pleasing images, we ensure that the generated first frames not only serve as functional inputs but also exhibit strong visual expressiveness. *2)* **image-text consistency scoring** evaluates how well an image aligns with its corresponding image prompt. This is achieved through an MLLM, which acts as a judge to assess whether the main subject, the implied action or transformation, and the surrounding scene match the textual description. The evaluation is conducted on a five-point scale, where a higher score indicates stronger alignment between the visual content and the intended VFX semantics. For efficiency and modularity, the same MLLM is employed across the entire AutoVFX pipeline to enhance reusability of the intelligent component.

## 2.4 VIDEO PRODUCER: AUTOMATIC VIDEO GENERATION AND EVALUATION

As the executor of video synthesis within the AutoVFX, the **Video Producer** is responsible for transforming static visual designs into fully realized VFX videos. Much like a producer in a film production pipeline, this role integrates the creative intent refined by the VFX Designer and the visual foundation established by the Scene Artist, while ensuring that the resulting videos are both technically sound and faithful to the envisioned effects.

**Image-to-Video Model Selection.** While the first-frame image sets the visual tone for the VFX generation, it is the video generation model that truly brings the visual effects to life. The Image-to-Video (I2V) model plays a crucial role in transforming static images into dynamic sequences, and its performance is directly tied to the quality of the VFX data. A well-chosen I2V model ensures that the visual effects are seamlessly integrated into the video, maintaining both aesthetic quality and consistency with the intended transformations. Several advanced I2V models have been developed in recent years, such as HunyuanVideo (Kong et al., 2024), KLING (Ding et al., 2025), Sora (Peng et al., 2025), and Wan (Wan et al., 2025). Among these, Wan2.2-I2V stands out due to its exceptional performance in generating high-quality videos with smooth transitions and strong semantic alignment with input prompts. These features ensure that the generated video remains faithful to the original concept while maintaining visual coherence across the entire sequence. Therefore, we have selected Wan2.2-I2V-A14B as the core model for the video producer, ensuring high-quality video generation with stable visual effects and precise video synthesis. Furthermore, Wan2.2 family supports not only I2V generation but also Text-to-Video (T2V) generation and First-Last-Frame-to-Video (FLF2V) generation, which meets the demands of AutoVFX extensibility branches.

**VFX Video Creation.** Based on the ranking of first frames produced in the Scene Artist, the top-$n$ images are selected as anchors for video synthesis. These images are then randomly divided into training and testing subsets to ensure both diversity and objective evaluation. Leveraging the pretrained Wan2.2-I2V model, the Video Producer combines the selected first-frame images with the crafted VFX prompts to generate videos, which serve as the primary carriers of VFX within AutoVFX.

**Automatic Video Assessment.** The primary goal of our AutoVFX is to mine the VFX potential of pretrained video generation models through data-driven learning. The success of this process hinges on ensuring the high quality of generated videos, which is why we have designed an automatic video evaluation module. This evaluation module assesses the effectiveness of the VFX from two key perspectives: *1) video quality* evaluated by VTSS (Wang et al., 2025) and FineVQ(Duan et al., 2025); *2) consistency* between the video content and the visual effects description. Specifically, we reuse the MLLM (Bai et al., 2025) from the image evaluation module, which evaluates how well the video matches the given textual description, considering the theme, motion description and environment details. The evaluation assigns a score based on the degree of alignment between the video and the description, with a scale from 1 to 5, where 5 indicates perfect consistency and 1 indicates no match. To compute the final score, the consistency score is first mapped to a percentage scale, and then combined with the video quality score through a weighted sum to form the overall video evaluation score (see Appendix D in Appendix).

## 2.5 VFX REFINER: TAILORED I2V MODEL CYCLE-FINETUNING

As the culmination of our AutoVFX model, the **VFX Refiner** serves as the final stage where the pretrained I2V model is transformed into a stable generator of high-quality VFX. By consolidating the outputs of the preceding stages, curating reliable training data, and applying cycle-based finetuning, it is dedicated to mining the latent VFX potential of the pretrained I2V model, thereby ensuring its ability to produce stable and high-quality visual effects.

**Curated Video Data Foundation.** The core of AutoVFX is to mine the VFX potential of the pretrained I2V models by constructing a pool of candidate VFX videos from themselves as training data. The effectiveness of mining is highly dependent on the quality of the training set, making it essential to rigorously curate the data before engaging in iterative refinement. Therefore, we employ the overall evaluation scores produced in the previous stage to filter the VFX videos, considering

only those above a predefined threshold as qualified training data. For each training round, the curated dataset is normalized to exactly the top-$k$ videos, with high-scoring samples repeated if insufficient and truncated if exceeding $k$. This design drives a steady improvement in training-data quality over successive cycle-finetuning iterations, thereby enhancing the progressive mining of VFX potential. In addition, we adopt an adaptive threshold strategy: for difficult VFX where initial videos fail to meet the quality standard, the threshold is lowered to reattempt selection. If no qualified samples are obtained, the AutoVFX recommends restarting the VFX generation procedure.

**Sustained Self-Mining.** The VFX Refiner iteratively mines the VFX potential of pretrained I2V models through a cycle-finetuning strategy. After each round of finetuning with curated high-quality training data, the updated I2V model is passed back to the Video Producer to generate a new batch of VFX videos. From these, the VFX Refiner selects the higher-quality samples to serve as the training data for the next round of finetuning. This iterative process forms a self-mining loop, in which the model's ability to capture and reproduce complex VFX is progressively reinforced. Empirical observations indicate that within two to three cycles, this strategy effectively saturates the model's VFX potential, yielding stable and high-quality results.

## 2.6 VFX-Bench Construction

To systematically evaluate the capability of video generation models in handling visual effects, we construct a dedicated VFX benchmark by curating a set of representative cases. The selected VFX cover diverse aspects including subjects, motion patterns, stylistic variations, and fantastical transformations. For clarity, each case is summarized in a concise subject–action form, while the complete set of VFX descriptions is provided in the Appendix. This design ensures coverage across a broad range of subjects, motion patterns, and stylistic variations, providing a challenging and representative benchmark for evaluation.

For evaluation metrics, we adopt the automatic VFX video evaluation framework introduced in Sec. 2.4. Specifically, two dimensions are assessed: *1)* **For the video quality**, we employ FineVQ (Duan et al., 2025), a perceptual video quality model trained on large-scale human preference data to provide reliable predictions. *2)* **For the VFX consistency**, we adopt a multimodal large language model with prompt designs that define five levels of consistency, ranging from perfect match to complete mismatch, thereby enabling standardized and interpretable evaluation. Based on these two dimensions, we compute a **Comprehensive Score** (CS) by taking the weighted average of the video quality score and the VFX-text consistency score. Specifically, for the consistency levels rated by MLLMs on a 1–5 scale, we map them to the percentage scale used for video quality (corresponding to 60–100). After applying the weighted averaging scheme, the resulting Comprehensive Score ranges from 30 to 100. In addition, we report a **Success Rate** (SR), which is defined based on the Comprehensive Score. According to human preferences, we set a reasonable threshold, and any generated VFX with a score above this threshold is regarded as successfully generated. The Success Rate is then computed as the proportion of successful VFX across the entire evaluation set, providing a straightforward indicator of the reliability of video generation models.

## 3 Experiments

### 3.1 Experimental Setup

In this section, we conduct comparative experiments with Wan2.2 (Wan et al., 2025) and Omni-Effects (Mao et al., 2025). Wan2.2 represents the state-of-the-art among general video generation foundation models, and using it as a baseline allows us to highlight better the value of our AutoVFX framework in mining VFX potential. Omni-Effects is one of the rare VFX-tailored generation models, which relies on fine-tuning with high-cost, manually curated VFX datasets to obtain strong VFX generation capability. Comparing with Omni-Effects not only demonstrates the advantage of our automated pipeline in terms of low-cost, but also demonstrates the superior performance of our method, especially in terms of generalization.

**Implementation Details.** Our proposed AutoVFX framework is designed with extensibility in mind, supporting rapid substitution of video generation models, LLMs, or MLLMs with stronger

Table 1: **Performance comparison with video generation models on VFX-Bench.** Our proposed AutoVFX boasts distinct advantages across the entire VFX-Bench. Here, **CS** and **SR** are metrics where larger values indicate better performance. When **SR** = 0, none of the VFX in the test set meet the threshold, whereas when **SR** = 1, all of them meet it.

| Metrics | Models | Ppl-Hug | Bldg-Launch | Car-Robot | Char-Anime | Flwr-Bloom | Char-Baby | Food-Dance | Ppl-Soar | Anim-Skate | Char-Jelly | Average |
|---|---|---|---|---|---|---|---|---|---|---|---|---|
| CS↑ | Wan2.2-I2V | 80.35 | 77.45 | 78.30 | 72.77 | 74.26 | 74.54 | 72.28 | 75.08 | 76.30 | 67.90 | 74.92 |
| | Round 1 | 80.40 | 78.64 | 78.29 | 75.91 | 78.63 | 75.13 | 77.53 | 77.88 | 76.33 | **71.18** | 76.99 |
| | Round 2 | **80.88** | 78.71 | **79.09** | 76.18 | **80.34** | 74.72 | **79.24** | 78.24 | 76.44 | 70.59 | **77.44** |
| | Round 3 | 80.65 | **79.07** | 78.63 | **76.39** | 80.14 | **75.39** | 78.78 | 78.05 | **76.50** | 69.23 | 77.28 |
| SR↑ | Wan2.2-I2V | **1.00** | 0.75 | **1.00** | 0.25 | 0.50 | 0.50 | 0.40 | 0.50 | 0.45 | 0.10 | 0.55 |
| | Round 1 | **1.00** | **0.90** | 0.95 | 0.80 | 0.90 | 0.55 | 0.75 | 0.80 | 0.50 | 0.15 | 0.73 |
| | Round 2 | **1.00** | **0.90** | **1.00** | **0.80** | **1.00** | 0.50 | **1.00** | **0.85** | 0.50 | **0.20** | **0.78** |
| | Round 3 | **1.00** | **0.90** | **1.00** | 0.75 | **1.00** | **0.65** | **1.00** | 0.80 | **0.60** | 0.05 | **0.78** |

Table 2: **Performance comparison with VFX-Tailored models on the VFX-Bench.** Here, instead of using the full VFX-bench, we select four categories of visual effects that are compatible with those supported in the Omni-Effects library.

| Metrics | Models | Bldg-Launch | Flwr-Bloom | Char-Baby | Ppl-Soar | Average |
|---|---|---|---|---|---|---|
| CS↑ | Omni-Effects (Mao et al., 2025) | 77.29 | 78.75 | 65.76 | 61.18 | 70.75 |
| | Wan2.2-I2V (Wan et al., 2025) | 77.45 | 74.26 | 74.54 | 75.08 | 75.33 |
| | Ours (Round 3) | **79.07** | **80.14** | **75.39** | **78.05** | **78.16** |
| SR↑ | Omni-Effects (Mao et al., 2025) | 0.85 | 1.00 | 0.00 | 0.00 | 0.46 |
| | Wan2.2-I2V (Wan et al., 2025) | 0.75 | 0.50 | 0.50 | 0.50 | 0.56 |
| | Ours (Round 3) | **0.90** | **1.00** | **0.65** | **0.80** | **0.84** |

alternatives as they become available. In this section, Wan2.2 (Wan et al., 2025) is adopted as the video generation backbone, Qwen3 (Yang et al., 2025) is chosen for prompt reasoning, and Qwen2.5-VL (Bai et al., 2025) is employed for consistency evaluation. For model configurations, the T2I model is set to generate images at 480P resolution, while the I2V model produces videos of 81 frames at the same resolution. In the Scene Artist stage, we generate a total of 500 first-frame candidates, from which the top 150 are selected according to the image evaluation module. These are further split into 100 training and 50 testing images. For each round of video generation, 100 videos are produced, and the top 40, ranked by the video evaluation module, are selected as training data for the VFX Refiner stage. Cycle-finetuning is typically performed for 2∼3 iterations, which we find sufficient to fully mine the VFX potential of the pretrained I2V model. All experiments are conducted on 8 H20 GPUs. During each finetuning round, a single video clip is repeated 10 times to stabilize optimization and ensure effective learning from limited high-quality data.

## 3.2 QUANTITATIVE RESULTS

**Comparison with video generation models.** As shown in Tab. 1, compared with the current state-of-the-art pretrained video generation model Wan2.2, our AutoVFX achieves consistent improvements in VFX generation across the diverse cases included in the VFX-Bench after up to three iterations of finetuning. The gains are particularly pronounced for those VFX where Wan2.2 initially performs poorly, demonstrating the effectiveness of AutoVFX in mining the latent VFX potential of pretrained models. Moreover, Tab. 1 further indicates that two to three cycles are generally sufficient to fully exploit this potential, while additional rounds yield diminishing returns. When uniformly evaluating with the model obtained after the third cycle, we observe that the **Average Comprehensive Score** on the VFX-Bench increases by **2.36**, and the generation **Success Rate** under the threshold of 75 improves by **23**%. Notably, for the challenging effect "Char-Jelly" which turns the character into jelly, the baseline performance of Wan2.2 is particularly poor, making it difficult for us to exploit stronger capabilities from this case.

**Comparison with VFX-Tailored models.** To further highlight the advantages of AutoVFX in specific VFX generation, we conduct a comparison with Omni-Effects (Mao et al., 2025). Since Omni-Effects does not fully support all visual effects in our benchmark, we select four representative cases that are similar to the categories available in its visual effects library. The evaluation metrics, shown in Tab. 2, demonstrate that the performance of VFX-tailored models is inherently constrained by the scope of their training datasets, leading to limited generalization. In some difficult cases, Omni-Effects even performs worse than pretrained video generation models, reflecting its limitations in generalization. By contrast, AutoVFX not only offers the advantage of a fully au-

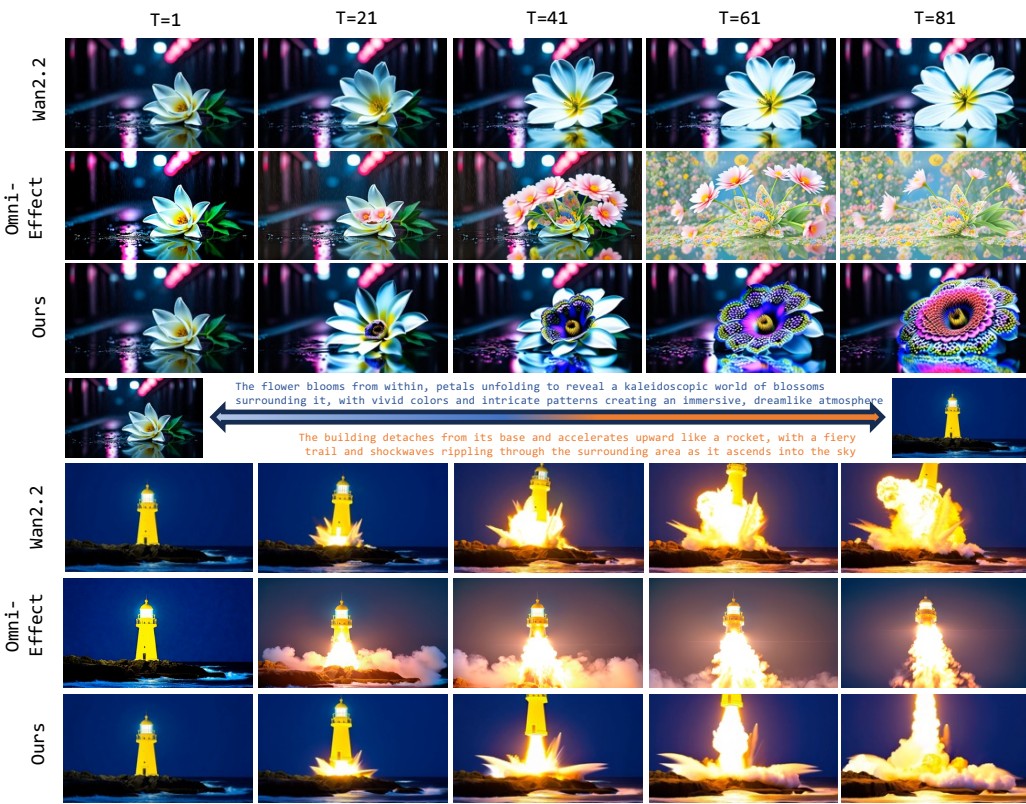

Figure 3: **Qualitative Comparison with Wan2.2 Wan et al. (2025) and Omni-Effects Mao et al. (2025).** Here we select two visual effects from the intersection of our VFX-Bench and the Omni-Effects library: "The flower blooms from within, unfolding into a kaleidoscopic world of blossoms" and "The building launches into the sky like a rocket".

tomatic procedure but also enhances the VFX capability of pretrained models, leading to generation performance that surpasses VFX-tailored models.

### 3.3 QUALITATIVE RESULTS

Considering that Omni-Effects is constrained by the limited scope of its supported visual effects library, we select two representative effects from the intersection with our diverse VFX-Bench for qualitative comparison: "The building launches into the sky like a rocket" and "The flower blooms from within, unfolding into a kaleidoscopic world of blossoms." As shown in Fig. 3, the results reveal distinct differences across models. For the pretrained Wan2.2, the outputs mainly focus on general video content generation, without adequately capturing the intended VFX-related magical effects. In contrast, Omni-Effects demonstrates its VFX-tailored advantages, which stem from high-quality training data but are inherently restricted by the limited coverage of its dataset. By comparison, our proposed AutoVFX leverages a fully automatic agent-based procedure to generate user-customized visual effects with low-cost and high efficiency, effectively mining the potential of pretrained video generation models. Moreover, AutoVFX can stably produce VFX that align more closely with user expectations.

### 3.4 ABLATION STUDIES

To examine the effectiveness of the key module designs for some roles in AutoVFX, we conduct ablation studies on specific VFX cases, as illustrated in Tab. 3. Since the ablation of certain modules

may affect the interactions among different roles within the agent, we detach the modules to be ablated from the agent and execute the ablated modules independently.

**Ablation of Image evaluation module in Scene Artist.** For the **Scene Artist**, the image evaluation module is responsible for controlling the quality of the first-frame, thereby influencing the quality of subsequent VFX generation. In this section, we conduct an ablation study on this module: instead of ranking the candidate images and selecting the top-$n$ as the first-frames, we randomly select $n$ images and pass them to the following procedure. The results show that removing the image evaluation module leads to a slight degradation in the final performance of the VFX generation model, demonstrating the effectiveness of the module design.

Table 3: **Quantitative Results of Ablating Partial Roles in the Visual Effect Agent.** This ablation study is uniformly conducted on the representative "Bldg-Launch" from the VFX-Bench.

| Scene Artist | | Video Producer | | CS | RS |
|---|---|---|---|---|---|
| Random | Rank | Flat | Stratified | | |
| ✔ | ✗ | ✗ | ✔ | 79.13 | 0.95 |
| ✗ | ✔ | ✔ | ✗ | 78.20 | 0.85 |
| ✗ | ✔ | ✗ | ✔ | **79.33** | **1.00** |

**Ablation of Stratified sampling in Video Producer.** The quality of generated videos requires even stricter control, so we further ablate the strategy used for selecting the top-$k$ videos. In our cycle-finetuning design, videos generated in each round are merged before selecting the top-$k$. Here, we compare *flat selection* and *stratified selection*: the former ranks all videos from different rounds together and directly selects the top-$k$, while the latter first compares videos of the same category across different rounds, selects the best among them, and then includes these candidates in the final ranking. The results show that stratified selection yields clearly superior performance.

**Ablation of Cycle-finetuning in VFX Refiner.** In the **VFX Refiner**, we introduce the cycle-finetuning strategy to ensure that AutoVFX can fully exploit the VFX potential of pretrained video generation models. As shown in Tab. 1, the finetuned models generally achieve their best performance in the 2nd or 3rd round. Additional qualitative results can be found in Appendix D.

## 4 CONCLUSION

We presented AutoVFX, an automated agent framework that mines the latent visual-effects capability of pretrained image-to-video models by closing the loop between prompt design, first-frame synthesis, video generation, multimodal evaluation, and cycle-finetuning. By replacing costly manual dataset creation with a lightweight, role-based agent and a scalable multimodal quality controller, AutoVFX produces high-fidelity, semantically consistent VFX videos at greatly reduced human cost. Empirical results on our VFX-Bench demonstrate that the approach substantially raises comprehensive quality and success-rate metrics compared to off-the-shelf I2V models and surpasses existing VFX-tailored solutions while remaining far more scalable. Beyond practical gains, the cycle-finetuning mechanism steadily improves both data and model quality, enabling stable generation of complex effects within a few iterations.

**Limitation and future work.** This paper only takes Wan2.2-I2V as an example to verify the effectiveness of the method. In the future, it can be extended to more scales and different series of models. Meanwhile, VFX-Bench can also be expanded with more VFX categories and quantities for large-scale evaluation.

## ETHICS STATEMENT

We have ensured that our study and dataset construction follow ethical standards, with no direct involvement of human subjects, and no foreseeable risk of harm. Data usage complies with privacy and legal requirements, and we have aimed to mitigate potential biases in annotations and model evaluation. We disclose no conflicts of interest or sponsorship that could influence the results.

## REPRODUCIBILITY STATEMENT

We have already elaborated on all the models or algorithms proposed, experimental configurations, and benchmarks used in the experiments in the main body or appendix of this paper. Furthermore, we declare that the entire code used in this work will be released after acceptance.

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

## OVERVIEW

The supplementary material presents more detailed descriptions of our method and more intuitive results of AutoVFX, to facilitate understanding and reproducibility:

- **Appendix A** provides an overview of related works.
- **Appendix B** provides the prompts used for LLMs and MLLMs, enabling reproducibility of our approach.
- **Appendix C** offers additional qualitative comparisons on the VFX benchmark between our method and other state-of-the-art models.
- **Appendix D** presents supplementary qualitative results from ablation studies, further validating the effectiveness of different modules in the visual effect agent.
- **Appendix E** presents supplementary qualitative results on real images.

## THE USE OF LARGE LANGUAGE MODELS

We use large language models solely for polishing our writing, and we have conducted a careful check, taking full responsibility for all content in this work.

## A    RELATED WORKS

**Video Generation Models.**   Initial progress in video generation stemmed from GAN  (Goodfellow et al., 2014) and VAE-based  (Kingma & Welling, 2013) architectures, which demonstrated the feasibility of generative modeling in the temporal domain but struggled to scale toward high-quality, temporally coherent outputs.  The advancement of diffusion models marked a paradigm shift in video generation, with AnimateDiff  (Guo et al., 2023) enabling plug-and-play temporal extensions of Stable Diffusion and Stable Video Diffusion (SVD) (Blattmann et al., 2023) establishing a systematic training recipe from text-to-image pretraining to large-scale video finetuning.  Recent architectural advances, particularly transformer-based backbones such as MMDiT, further enhanced semantic alignment and controllability, laying the foundation for large-scale video foundation models. The text-to-video line, proprietary systems like OpenAI Sora and Google Veo 3 (Google Deep-Mind) demonstrated long-horizon, physically consistent generation, while open-source counterparts including CogVideoX  (Yang et al., 2024), HunyuanVideo  (Kong et al., 2024), Open-Sora  (Zheng et al., 2024), and VideoCrafter1  (Chen et al., 2023) provided reproducible pipelines for the research community.  In parallel, image-to-video generation advanced through both commercial products such as Runway Gen-3/4 and Pika 1.0, and open models like SVD  (Blattmann et al., 2023) and VideoCrafter1  (Chen et al., 2023).  Most recently, Wan2.1 and Wan2.2  (Wan et al., 2025) introduced mixture-of-expert training and multitask generation capabilities, pushing the frontier of high-resolution and temporally coherent generation, and making VFX applications increasingly feasible in practice.

**Visual Effects (VFX) Generation.**   Image-to-Video (I2V) generation has emerged as a natural entry point for synthesizing visual effects, as it provides a strong conditional anchor for controlling spatial layout and scene composition.  However, open-source I2V procedures often struggle to render complex VFX with sufficient fidelity and temporal coherence, while closed-source systems, although achieving higher perceptual quality, are still being costly, proprietary, and difficult to extend with user-specified effects.  Recent attempts have explored more controllable solutions. VFX Creator  (Liu et al., 2025b) integrates video transformers with spatial–temporal adapters and introduces the Open-VFX benchmark, but it relies heavily on curated data and covers only a limited range of categories, limiting generalization. Similarly, Omni-Effects  (Mao et al., 2025) unifies multiple VFX types via LoRA-MoE and spatial-aware prompts, yet constructing its Omni-VFX dataset requires substantial manual effort and domain expertise, making scalability and cross-domain applicability challenging. Editing-style systems such as AutoVFX  (Hsu et al., 2025) couple scene modeling with physical simulation but incur heavy engineering overhead.  These limitations motivate a data-free and automated alternative: we propose VFX-Agent, a multi-agent pipeline that mines the latent VFX potential of pre-trained video generators—automating prompt design, frame

and video screening, and self-play training—to deliver stable, high-quality effects at low marginal cost, enabling scalable VFX creation for downstream applications.

**Agentic Video Content Creation and Evaluation.** Recent progress in large language models has sparked a growing body of research on agentic systems for multimodal content creation. Representative works such as GenArtist (Wang et al., 2024), PresentAgent (Shi et al., 2025), Paper2Poster (Pang et al., 2025) and PodAgent (Xiao et al., 2025) highlight how role-specialized agents can collaborate to produce complex creative artifacts with minimal human intervention. Extending this paradigm to video synthesis, VFX generation presents an ideal use case: diverse roles such as "VFX director" or "concept artist" can be instantiated as autonomous agents, thereby addressing the high cost, labor intensity, and scalability limitations of conventional VFX procedures. A complementary challenge lies in automatic evaluation, which is critical for closing the loop of data-free VFX generation. Performing in LAION (Schuhmann, 2022), video benchmarking toolkits such as VBench (Huang et al., 2024) and VEBench (Sun et al., 2025), and perceptual quality metrics such as Koala-36M (Wang et al., 2025) and FineVQ (Duan et al., 2025). While these methods provide insights into general video fidelity, semantic consistency, and perceptual alignment, they remain insufficient for VFX-specific assessment, where style controllability and visual plausibility are paramount. To address this gap, we design a fully automated evaluation pipeline that spans from image-level to video-level filtering, augmented by Multimodal Large Language Models (MLLMs) for contextual judgment. This integration enables robust self-mining of high-quality VFX data, ultimately supporting our goal of automated, scalable VFX generation.

## B  PROMPTS FOR LLMS AND MLLMS

To ensure reproducibility, we provide the exact prompts used for both LLMs and MLLMs in our framework. For LLMs, we design two types of prompts: (i) prompts for refining VFX descriptions, and (ii) prompts for generating corresponding first-frame descriptions. For MLLMs, we design prompts for multi-modal evaluation. Since the image and video evaluation prompts share highly similar structures, we only include the prompt for video evaluation here. The detailed prompts are shown below.

**LLM Prompt: VFX Description Refinement.**

> You are a creator of an AI model for generating videos from images. There is a powerful basic video model that can generate videos from images with prompts, and you want to explore its potential capabilities and novel playstyles in the application of special effects video generation.
>
> Your task is to rewrite the simple, casual video special-effect description into a high-quality video task prompt, which include three parts:
>
> 1. Theme — the main subject of the video (e.g., person, animal, vehicle, building, natural phenomenon, etc.).
>
> 2. Motion description — what happens in the video (e.g., running, transforming, exploding, glowing, dissolving, etc.).
>
> 3. Scene or motion detail description — the environment, context, atmosphere, or details that make the motion vivid and cinematic.
>
> Rules:
>
> - Preserve the subject exactly as given in the input (do not replace it with synonyms or more specific/general terms).
>
> - Preserve the core action/effect exactly (do not change the meaning of the motion).
>
> - Do not add new environments, atmospheres, or objects unless they are already implied by the input.
>
> - Output must stay focused on the subject and the effect, written in one clear sentence.
>
> Examples:
>
> - Input: "Two people hug each other." Output: "Two people hug each other, their full bodies clearly visible so the effect can be seen without obstruction."

- Input: "A building is flying into the sky." Output: "The building detaches from its foundation and launches upward into the sky, with smoke and debris erupting from the city streets below."

Now, rewrite the following video task prompt into a detailed and standardized form:

Input video special-effect description: {}

Output format (strictly follow this style, do not include any explanations or extra text):

prompt1: `<rewritten video task prompt>`.

**LLM Prompt: First-Frame Description Generation.**

You are designing prompts for the first frame of a special-effects video. The video task is described as: {}

Your job is to write 100 different prompts that describe what the very first frame of this video looks like. The first frame is a static picture — nothing is moving yet. It should capture the subject of the video and the setting before the effect begins.

Strict rules:

- Every prompt must clearly include the main subject described in the video task (e.g., if the task involves a building, every prompt must show that building; if it's a person, the person must appear).

- You may vary the environment, background, lighting, weather, season, or atmosphere to create diversity, but the subject must remain the clear focus.

- Do not introduce details or objects unrelated to the video task. Stay consistent with the theme.

- The variations should always be compatible with the described video effect, so the first frame can naturally lead into the transformation.

- Each prompt must describe a still image only — no actions, no transitions.

- Do not repeat prompts; all 100 must be unique.

- Write each prompt as one clear English sentence, 18–30 words long.

Examples:

Video task: "Two people face the camera, then turn around and hug each other affectionately."

Possible first frame prompts:

- "Two people stand together in a warmly lit living room, both looking directly at the camera."

- "A young boy and girl stand in the middle of a playground, facing forward with their full bodies visible."

Output format (follow strictly, no extra text):

prompt1: `<sentence>`.

prompt2: `<sentence>`.

...

prompt100: `<sentence>`.

**MLLM Prompt: Video Evaluation.**

You are a reviewer of video-to-text alignment. Your task is to judge how well a video matches a given text description, considering three aspects:

- theme (the main subject of the video),

- motion description (what happens, how it changes or moves),

- scene/motion detail description (where it happens, visual conditions, environment, or details of the motion).

Focus on whether the subject and the main motion clearly align with the description. Rate the consistency on a scale from 1 to 5 with the following standards:

- **[score: 5]** Perfect match. The main subject, motion, and scene all strongly align with the description. The effect or transformation is fully recognizable and faithful.
- **[score: 4]** Good match. The subject and motion align well, but there are small omissions, timing issues, or minor mismatches in details. The overall intent remains very clear.
- **[score: 3]** Partial match. The subject is correct, but the motion is vague, incomplete, or only loosely related. Some important details from the description may be missing or incorrect.
- **[score: 2]** Poor match. The subject or motion is significantly different from the description. The video only faintly resembles the intended idea.
- **[score: 1]** No match. The subject, motion, and scene do not correspond to the description at all. The meaning is completely inconsistent.

Strict output rules:
- Output exactly one line.
- The format must be: [score: `<number from 1 to 5>`]
- Do not output anything else.

## C  QUALITATIVE COMPARISONS ON THE VFX BENCHMARK

On our VFX benchmark, four out of the ten visual effects are similar to those supported by Omni-Effects. As shown in Fig. A1, we present qualitative comparisons among our method, Omni-Effects, and Wan2.2 on these four categories.

## D  SUPPLEMENTARY QUALITATIVE RESULTS OF ABLATION STUDIES

For the remaining six effects, which are not supported by Omni-Effects, we conduct qualitative comparisons only with Wan2.2, including results across multiple finetuning cycles, as illustrated in Fig. A2.

## E  QUALITATIVE COMPARISONS ON REAL IMAGES

In addition, we conduct qualitative comparisons of visual effects on real images as shown in Fig. A3. This experiment further demonstrates the applicability of AutoVFX beyond synthetic benchmarks, highlighting its ability to generate realistic and visually compelling VFX under practical scenarios.

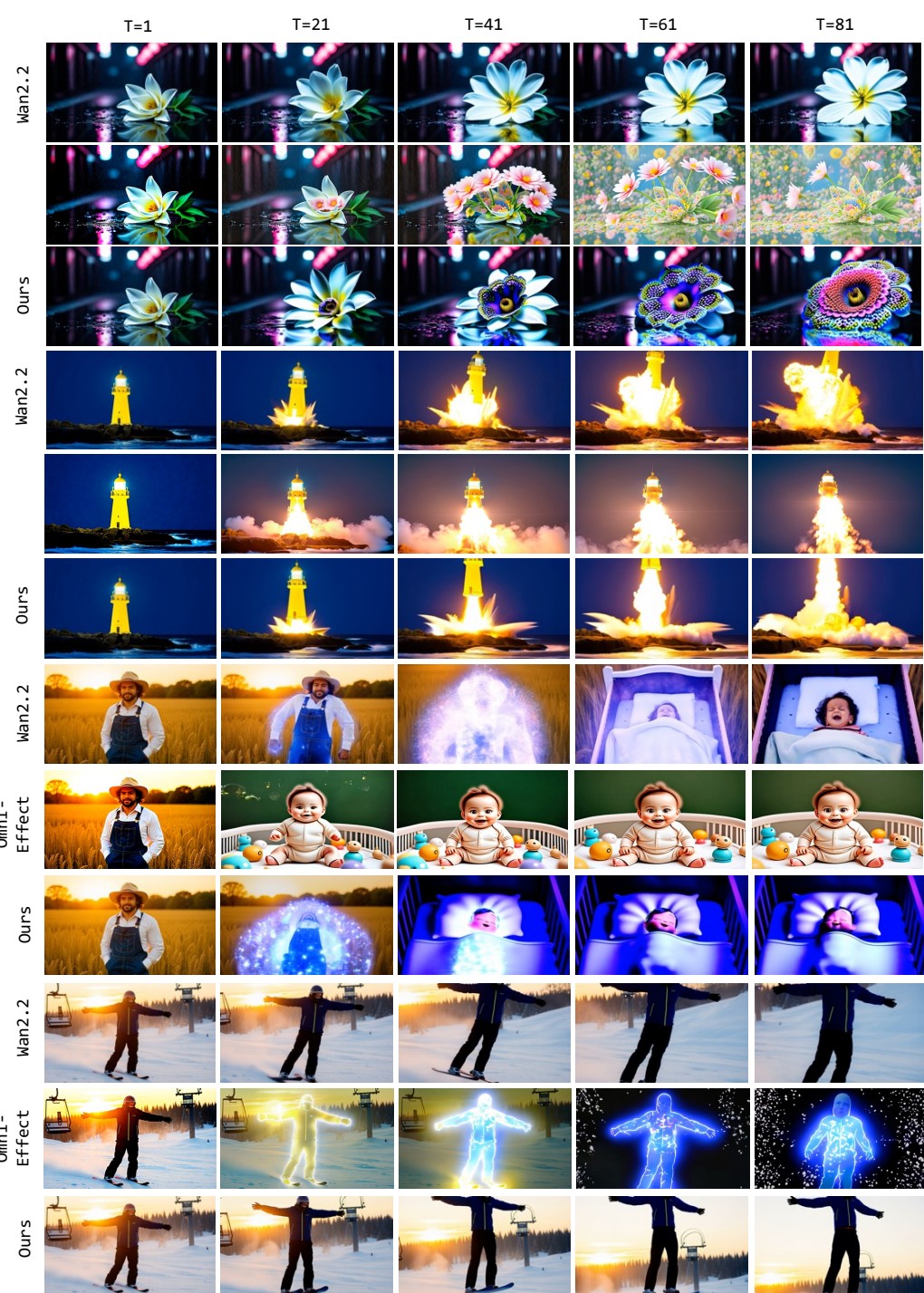

Figure A1: **Qualitative Comparison with Wan2.2 Wan et al. (2025) and Omni-Effects Mao et al. (2025).**

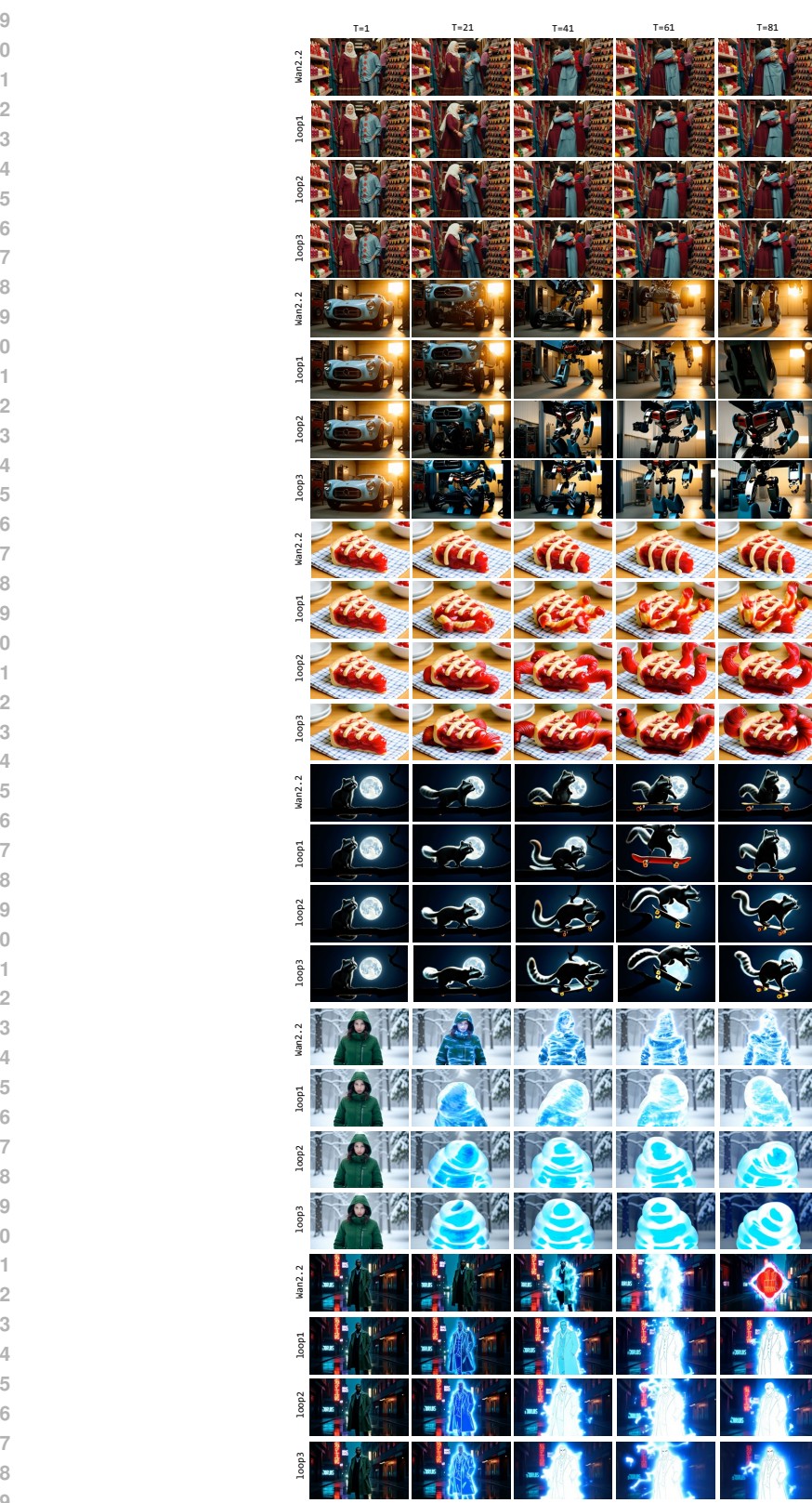

Figure A2: **Qualitative Comparison with Wan2.2 across Multiple Cycles.**

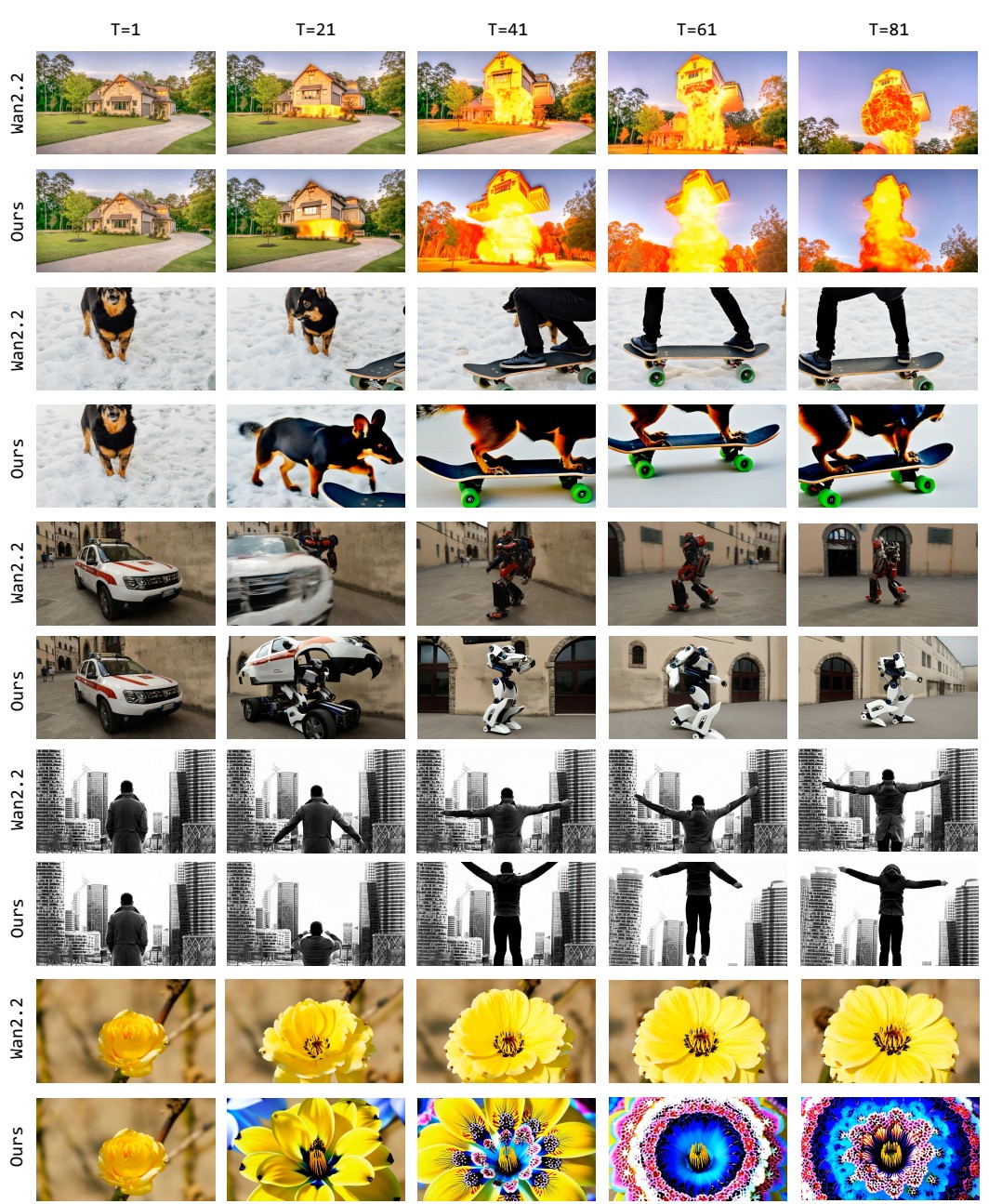

Figure A3: **Qualitative Comparison Comparisons on Real Images.**

