# OpenReview forum: "Data-free VFX Self-Mining"
_ICLR.cc/2026/Conference — ICLR 2026 Conference Withdrawn Submission_

### Official Review · Reviewer_4qtr · 2025-10-21

**Soundness:** 2
**Presentation:** 2
**Contribution:** 2
**Rating:** 2
**Confidence:** 5

**Summary:**

This paper introduces an agent system optimized for automated VFX enhancement. By designing and filtering VFX-related prompts and videos, the system iteratively fine-tunes existing video generation models, continuously improving the VFX quality of the generated videos. Additionally, the authors propose a new benchmark, VFX-Bench, which evaluates generated videos from the perspectives of video quality and VFX consistency. The method achieves state-of-the-art results on this benchmark

**Strengths:**

1. An agent system that can automatically improve the performance of VFX without real world data collection.
2. The result shows good VFX.

**Weaknesses:**

Compared to the original Wan2.2, the video quality after fine-tuning with the proposed method has clearly and significantly deteriorated. In all the visualizations provided by the authors, both the main subjects and backgrounds become noticeably blurred (i.e., cartoon style, lack detail), indicating that enhancing specific VFX features severely compromises overall video quality. However, this issue is not reflected at all in the benchmark introduced by the authors, which raises concerns about the accuracy and reliability of the benchmark itself. This also highlights substantial flaws in the automated fine-tuning process. As mentioned in the authors' motivation, existing i2v models possess a certain capability for VFX generation, but is it possible that the original video quality is inherently low? If so, these may not truly represent ideal VFX data.

I am unsure how the authors conducted the video quality evaluation, but based on the visualizations provided in the paper, none of the conclusions appear to be reliable.

**Questions:**

See the Weaknesses.

---

### Official Review · Reviewer_Nrrt · 2025-11-01

**Soundness:** 3
**Presentation:** 3
**Contribution:** 2
**Rating:** 6
**Confidence:** 3

**Summary:**

This paper introduces AutoVFX, an automated framework that extracts and improves VFX capabilities from existing video generation models without needing new data. The system uses a closed-loop agent to generate, filter, and then use the best video clips to iteratively fine-tune the model, progressively enhancing its performance on VFX tasks.

**Strengths:**

- The core idea of "self-mining" is practical. Instead of the expensive process of creating new datasets, the paper proposes a closed-loop system that automatically generates its own training data to improve an existing model. This is somewhat similar to manually RL
- The commitment to release all data (VFX-Bench) and code is good.
- The introduction of VFX-Bench and the associated evaluation metrics (Comprehensive Score and Success Rate) is a valuable contribution. This provides a structured way for the community to measure and compare performance on VFX tasks, which has been lacking.

**Weaknesses:**

- There seems to be a disconnect between the quantitative metrics and the qualitative results. While the scores show significant improvements over the Wan2.2 baseline, some of the video comparisons in the supp show only subtle visual improvements, which makes it hard to fully appreciate the real-world impact of the reported metric gains.
- The experiments are limited to a single base model (Wan2.2-I2V). While the results are promising, it's unclear how well this self-mining framework would generalize to other video generation models with different architectures or initial capabilities. The paper's claims of general applicability would be much stronger if tested on at least one other distinct model family.
- The success of the framework seems highly dependent on the initial "latent capability" of the base model. If a model is completely unable to produce even a few decent examples for a complex VFX task (like the "Char-Jelly" case mentioned), the self-improvement loop has no high-quality data to start with and the system fails. This suggests the method might not work for truly novel effects that are far outside the base model's training distribution.

**Questions:**

- The automatic video assessment relies heavily on an MLLM to score text-video consistency on a 1-5 scale. Could you clarify how the reliability of this MLLM judge was validated? For instance, were its scores compared against human judgments on a subset of the data to ensure it aligns with human perception of quality?
- The cycle-finetuning process needs more detail. The paper mentions that in each round, the top 40 videos are selected and each clip is repeated 10 times for fine-tuning. Could you elaborate on why this repetition is necessary and how you prevent the model from simply overfitting to this small, repetitive set of high-quality examples?

---

### Official Review · Reviewer_Q2Ta · 2025-11-01

**Soundness:** 3
**Presentation:** 3
**Contribution:** 3
**Rating:** 4
**Confidence:** 4

**Summary:**

The paper introduces AutoVFX, a closed-loop multi-role agent that first designs VFX prompts using an LLM as the VFX Designer, then synthesizes and filters first frames using T2I as the Scene Artist, runs I2V generation and automatic video-level filtering as the Video Producer, and finally cycle-finetunes the I2V backbone on its own filtered outputs as the VFX Refiner. It also presents VFX-Bench with two aggregate metrics: Comprehensive Score and Success Rate, which combine perceptual video quality from FineVQ and VTSS with an MLLM-based VFX-text consistency score. Experiments show improvements over Wan2.2-I2V and a partial comparison with Omni-Effects. Two to three refinement cycles are reported to saturate the gains.

**Strengths:**

S1: Cycle-finetuning using automatically selected top-k videos leads to consistent improvements in Comprehensive Score and Success Rate over 2 to 3 rounds, with diminishing returns after that. This shows that generated data can provide useful signal when filtered strictly.

S2: The four-role agent clearly separates creative prompt design, first-frame selection, video generation, and iterative finetuning. Each role is defined with specific model choices such as FLUX for T2I, Wan2.2-I2V, and Qwen2.5-VL as the judge, and the loop is straightforward to reproduce or extend.

**Weaknesses:**

W1: VFX-Bench uses the same MLLM family, Qwen2.5-VL, both to filter training data and to judge text–video consistency, and the Comprehensive Score includes this judge in its weighted sum. If the MLLM has biases such as style preferences or sensitivity to phrasing, training may optimize toward the judge rather than true VFX quality, creating a risk of evaluation circularity. There is no human audit or second-assessor check on the scores from VFX-Bench.

W2: The method does use data—specifically, self-generated clips from the backbone and LLM or MLLM outputs as supervision. Data-free only means that no external manually labeled VFX dataset is used. The paper should make this clear to avoid confusion about the origin of the supervision.

W3: For the proposed metric, the Comprehensive Score maps MLLM ratings from 1 to 5 into a 60 to 100 range and averages this with video-quality scores. The Success Rate counts samples with CS above a threshold as successful. Neither metric reflects artifact severity or temporal extent, such as brief versus sustained failures. Rankings may vary under different weights or thresholds. Confidence intervals and sensitivity analyses are not included.

W4: The ablation covers random versus ranked first-frame selection and flat versus stratified video selection on one case, Bldg-Launch. There is no ablation on the choice of MLLM, the weights in CS, the number of refinement cycles beyond three, the top-k selection size, or the prompts used by the judge.

**Questions:**

Both the training loop and the benchmark depend on Qwen2.5-VL for consistency scoring. How do you ensure that the model is not simply optimizing for this evaluator’s bias? Was any testing done using an independent assessor such as InternVL3, GPT-4V, or human raters?

Were any human studies conducted to check the correlation between the Comprehensive Score or Success Rate from VFX-Bench and perceived visual quality or physical realism? Without human alignment, how can you be sure that the reported gains reflect real visual improvement rather than overfitting to the judge?

How diverse are the effects included in VFX-Bench? Are there categories that depend strongly on compositing realism, such as smoke or water, compared to others that are mainly stylistic? How balanced is the dataset in terms of difficulty and domain coverage?

The Comprehensive Score combines several sub-scores using fixed weights. How sensitive are your results to this weighting? Have you run any ablations where the weight scheme or the Success Rate threshold is varied?


You mention that 2 to 3 cycles are sufficient. Did you try longer loops, such as 5 or more, to confirm convergence? What safeguards are in place to prevent over-optimization or a drop in quality?
Since the same LLM family is used to generate prompts and to evaluate them, could linguistic similarity lead to inflated text–video alignment scores? Did you test with paraphrased prompts or syntactic variations that were not seen during training?

---

### Official Review · Reviewer_Cffv · 2025-11-02

**Soundness:** 2
**Presentation:** 3
**Contribution:** 2
**Rating:** 2
**Confidence:** 4

**Summary:**

This paper presents AutoVFX, a framework for automated VFX video generation from pretrained I2V models. The system uses LLMs for prompt generation, T2I models for first-frame synthesis, and iterative supervised finetuning on automatically selected high-quality outputs. While the problem is interesting and the pipeline is well-engineered, the paper suffers from critical methodological issues including unvalidated metrics, lack of human evaluation, and overclaimed contributions.

**Strengths:**

- The paper addresses a real limitation of current I2V models with an automated, data-free approach that avoids expensive manual annotation.

- The four-module agent design is intuitive and well-motivated, with each component serving a clear purpose in the pipeline.

- VFX-Bench provides a useful resource for evaluating VFX generation capabilities with clearly defined metrics (CS and SR).

**Weaknesses:**

- The paper proposes metrics (CS, SR) claimed to follow "human preferences" but provides zero human validation studies. All quantitative results in Tables 1-3 are based entirely on automated metrics with no evidence these correlate with human perception. Without correlation studies showing CS/SR align with human judgments, we cannot trust any of the reported improvements or baseline comparisons.

- The proposed metod is also limited, There's no reinforcement learning, no reward modeling, no true bootstrapping—just iterative supervised learning with improving data quality.

- The paper misses citations to the entire self-training literature (Self-Instruct, Constitutional AI, RLAIF, STaR). More discussion with the related with the prior related work is needed.

**Questions:**

- What is the human corrlation with the proposed benchmark and metric.

- In table one, the "Char-Jelly" SR goes from 0.20 (Round 2) → 0.05 (Round 3). This is a massive drop. What's happening here?

- Training Hyperparameters and Reproducibility

---

### Note · Authors · 2025-11-12

I have read and agree with the venue's withdrawal policy on behalf of myself and my co-authors.